# Can resting heart rate explain the heart rate and parasympathetic responses during rest, exercise, and recovery?

**Giliard Lago Garcia**[1,2]*, **Luiz Guilherme Grossi Porto**[1,2], **Carlos Janssen Gomes da Cruz**[1,2], **Guilherme Eckhardt Molina**[1,2]

1 Exercise Physiology Laboratory of Faculty of Physical Education, University of Brasilia, Brasilia, Distrito Federal, Brazil, 2 Research Group in Physiology and Epidemiology of Exercise and Physical Activity (GEAFS)—University of Brasilia, Brasilia, Brazil

* giliardlgarcia@gmail.com

**Data Availability Statement:** All relevant data are within the paper and Supporting Information files.

**Funding:** This work was supported by Fundação De Apoio À Pesquisa Do Distrito Federal - FAPDF

## Abstract

The effect of resting heart rate (RHR) on the heart rate (HR) dynamics and parasympathetic modulation at rest, exercise, and recovery remain to be clarified. This study compares HR and parasympathetic responses at rest, during submaximal exercise testing and recovery in young, physically active men with different RHR average values. HR and parasympathetic responses were compared between two groups: bradycardia group–RHR < 60 bpm (BG, n = 20) and normocardia group–RHR $\geq$ 60 $\leq$ 100 bpm (NG, n = 20). A Polar RS800® was used to record the RR-interval series and HR at rest in the supine position, following the postural change (from supine to orthostatic position) and in the orthostatic position for 5 minutes, as well as during and after a submaximal exercise testing. Statistical analysis employed the MANOVA, Mann-Whitney, and Simple Linear regression test with a two-tailed p-value set at $\leq$ 0.05. BG at rest showed lower HR in the orthostatic position, higher parasympathetic activity in the supine and orthostatic positions, and higher parasympathetic reactivity than NG (p $\leq$ 0.01) after the postural change. BG before exercise showed lower HR and higher values of the chronotropic reserve and parasympathetic withdrawal than NG (p $\leq$ 0.01) throughout the exercise. Following the exercise, BG showed higher values of HR recovery (HRR) and parasympathetic reactivation in the 3rd and 5th minutes of recovery than NG (p $\leq$ 0.01). Lastly, the RHR can explain the variance of the HR at rest, during exercise testing, and recovery from 11 to 48%. We concluded that BG (RHR < 60 bpm) showed higher chronotropic and parasympathetic modulation at rest, higher chronotropic reserve, parasympathetic withdrawal during the submaximal exercise test, and faster HRR and parasympathetic reactivation after effort in young physically active men.

## Introduction

The chronotropic and cardiac autonomic responses assessed at rest, during exercise, and post-exercise recovery are considered powerful independent predictors of cardiovascular morbi-mortality in the general population [1–3].

Resting heart rate (RHR) is often used in the clinical setting to assess the relative strength of parasympathetic activity and sympathovagal balance in the heart [4]. During an incremental

under Grant 11/2022 and by Programa de Pós-graduação em Educação Física (PPGEF) under Grant Edital PPGEF n. 11/2022. The funders had no role in study design, data collection and analysis, decision to publish, or preparation of the manuscript.

**Competing interests:** The authors have declared that no competing interests exist.

exercise test, heart rate (HR) increases due to parasympathetic withdrawal and sympathetic activation [5]. In contrast, after exercise, the short-term post-exercise HR adaptation, the heart rate recovery (HRR), occurs in response to simultaneous rapid parasympathetic reactivation and progressive sympathetic deactivation [5]. Consequently, a high RHR, slow HR response during an exercise test, and a delayed HRR are associated with reduced parasympathetic activity, withdrawal impairment, and reactivation of parasympathetic activity, respectively [1–3, 6, 7].

Even though those physiological measurements mentioned above are considered independent predictors of risk for cardiovascular mortality [1–3], some studies have investigated the association between parasympathetic activity at rest, evaluated by heart rate variability (HRV), with HRR [8–16] by considering it to be a promising area of clinical and functional evaluation.

In this clinical setting, some studies have reported a significant association between resting parasympathetic activity with HRR after a maximal or submaximal incremental exercise test [9, 11, 14–16]. When evaluating the HR during the exercise, only one study showed a significantly positive association between parasympathetic activity at rest with chronotropic reserve (CR); the difference between HR at peak exercise and the RHR in the supine position, during the maximal exercise test [16]. On the other hand, other studies have not shown a significant association between parasympathetic activity at rest with HRR or parasympathetic activity after incremental exercise tests [8, 10, 12, 13].

Indeed, due to different approaches used in those studies above to evaluate resting status, inconsistent results may be produced because the chronotropic response and parasympathetic modulation are adaptative phenomena that are affected by body positions (supine, sitting, or standing), types of exercise test protocols (maximal or submaximal), and types of recovery protocols (active or passive) [5, 17].

So, the inconclusive results regarding the interaction between cardiac resting measurements and the chronotropic and parasympathetic responses during and after an exercise test remain, and new approaches are needed to expand prior investigations. To the best of our knowledge, no studies have shown information on the effect of different RHR average values on HR and parasympathetic activity at rest, during the exercise test, and during recovery.

Furthermore, the hypothesis that different RHR average values can affect HR and parasympathetic activity during and after an exercise test opens the possibility of a new approach (or analysis) using RHR average values, which may add helpful information as a preliminary tool for decision-making (i.e., stress management) for healthcare professionals bringing essential and complementary information related to individuals cardiac autonomic capacity without the expense of clinical exercise tests or maximal/near the maximal effort required for exercise and recovery analysis.

Therefore, we hypothesize that physically active young men with different RHR average values in the supine position show different HR and parasympathetic activity responses at rest, during, and after a submaximal exercise test.

Accordingly, our objectives were: (a) to compare the HR and parasympathetic responses during rest, exercise, and post submaximal exercise in physically active young men with different RHR average values; (b) to develop an explication regression based on the effect of RHR average values on the HR response during rest, exercise, and post submaximal exercise in physically active young men.

## Materials and methods

### Participants

We conducted a cross-sectional study, enrolling 40 young and physically active males with a median (quartiles) age of 26.7 (20.0–40.0) years and body mass index (BMI) equal to 24.2

(20.1–28.2) kg/m$^2$. Participants were eligible for inclusion if they were men, physically active ($\geq$ 150 min of moderate-vigorous physical activity per week, International Physical Activity Questionnaire—IPAQ) [18], non-athletes, healthy (no medical restrictions nor known disease), and aged between 20 and 40 years old. Volunteers underwent exercise testing 2h after breakfast, between 8:00 and 10:00 a.m., and were previously instructed to abstain from stimulants and alcoholic beverages and physical activity for at least 24 h before evaluation. The studies involving human participants were reviewed and approved by the Ethical Committee on Human Research of the University Center Euro-Americano (UNIEURO, approval number: 006/2011) in agreement with the Declaration of Helsinki. The participants provided their written informed consent to participate in this study, and each participant signed informed written consent.

## Study design

Initially, we collected clinical basic physiological data, anthropometrical measurements, and information on lifestyle habits. Afterward, in a quiet exercise physiology laboratory room, at a temperature between 22 to 24˚C and relative humidity of 50–60%, continuous HR was recorded according to a standardized protocol previously described to obtain the R-R interval series [15, 17]. First, a valid five-minute R-R interval series and HR were obtained following 10 minutes of rest in the supine position. After, participants were asked to actively adopt the orthostatic posture at the bedside. Two minutes after the postural change, the blood pressure was measured to verify the absence of significant postural hypotension, and an additional five minutes of the R-R interval series and HR were recorded.

The submaximal treadmill exercise test was applied immediately after recording the R-R interval series (at supine and orthostatic postures). Soon after the interruption of the exercise test, the participants proceeded to the post-exercise active orthostatic recovery at 2.4 km/h and 2.5% grade [17].

The subjects were allocated into two groups based on the supine RHR average values to test our hypothesis. Those who presented bradycardia at rest (RHR < 60 bpm) were grouped in the bradycardia group (BG n = 20), and those who presented an RHR average values within the normal range (HR $\geq$ 60 bpm < 100 bpm) were allocated to the normal HR group (NG n = 20).

## Heart rate and heart rate variability analysis

The HR and R-R intervals series were recorded using a valid and reliable heart rate monitor Polar$^{\circledR}$ (model, RS800CX, Polar ™, Kempele, Finland) with a sample rate of 1000Hz [16, 17]. Then, each R-R interval series file was transferred to a computer for offline data processing and analysis of HR and HRV of the R-R interval utilizing the Polar Pro Trainer 5 software and the Kubios HRV software (version 2.2, Kuopio, Finland), respectively [19].

All R-R segments were visually analyzed, and occasional artifacts were manually or automatically removed (< 1% of recording) [20]. The automated artifact identification and removal were performed using the threshold method, which consists of selecting R-R intervals that were larger or smaller than 0.45s (very low), 0.35s (low), 0.25s (medium), 0.15s (strong), or 0.05s (very strong) compared to average R-R intervals [19]. We used the medium threshold method that only removed the visually observed ectopic points, as long as the tracing did not lose the physiological pattern and the removal did not exceed 1% of the recording [19, 20].

The parasympathetic activity was evaluated by the square root of the square of successive adjacent R-R intervals difference (rMSSD), a time-domain index associated with respiratory sinus arrhythmia [21]. At rest, during, and after the submaximal treadmill exercise test,

rMSSD was assessed using two different metrics: the short-term HRV measurement, i.e., five minutes, and the ultra-short-term HRV measurement, i.e., $\leq$ one minute, that was analyzed at peak effort (30 seconds final) and in each segment of one minute throughout the active recovery phase, i.e., $1^{st}$, $3^{rd}$, and $5^{th}$ minute. These quantitative analysis methods do not require the stationarity of the HR series, allowing the analysis during and after the exercise [21, 22].

The HR at supine (RHR) and orthostatic ($HR_{ort}$) positions were recorded as previously described at resting conditions, and absolute ($\Delta_{abs}RHR$) and relative ($\Delta_{\%}RHR$) variations were calculated by subtracting $HR_{ort}$ from RHR. During the submaximal treadmill exercise test, HR recording was initiated immediately before starting the exercise test ($HR_{initial}$) and stopped when participants reached 85% of their maximum predicted HR ($HR_{peak}$) by the Tanaka formula [23]. The CR was calculated by subtracting $HR_{peak}$ from RHR [24, 25]. During the active recovery, HR was recorded at the $1^{st}$, $3^{rd}$, and $5^{th}$ minute, and the absolute (HRR) and relative (%HRR) values of HRR were calculated by subtracting HR at the $1^{st}$, $3^{rd}$, and $5^{th}$ minute during the recovery phase from the $HR_{peak}$ [26].

At rest, the rMSSD at supine ($rMSSD_{sup}$) and orthostatic ($rMSSD_{ort}$) positions were recorded, and the absolute ($\Delta_{abs}rMSSD_{rep}$) and the relative ($\Delta_{\%}rMSSD_{rep}$) variation values of rMSSD were calculated by subtracting $rMSSD_{ort}$ from $rMSSD_{sup}$. During the submaximal treadmill exercise test, the rMSSD was recorded when the participants reached 85% ($rMSSD_{peak}$), and we obtained the parasympathetic withdrawal by subtracting $rMSSD_{peak}$ from $rMSSD_{sup}$ through the absolute ($\Delta_{abs}rMSSD_{exer}$) and the relative ($\Delta_{\%}rMSSD_{exer}$) rMSSD variation. At recovery, the rMSSD was obtained at the $1^{st}$, $3^{rd}$, and $5^{th}$ minutes over the active recovery phase.

## Treadmill submaximal exercise testing

All participants performed the submaximal treadmill exercise testing on a conventional treadmill (Centurion–Micromed, Brazil). The use of submaximal exercise testing was by the risk level to the volunteers and the availability of appropriate equipment and personnel to determine the HR and HRV response to one determined submaximal work rate [27].

The submaximal exercise test started with two minutes of warm-up at a speed of 3.0 km/h and 2.5% grade. The grade remained constant throughout the test and during recovery. After the two minutes of warm-up, the submaximal exercise test protocol started at a speed of 4.0 km/h and 2.5% grade, and the speed was increased by 1.0 km/h every minute until participants reached $HR_{peak}$.

After achieving this submaximal intensity (85% of their predicted maximum HR), the exercise test was interrupted, and a 5-minute active recovery period was initiated with volunteers in standing positions at 2.4 km/h and 2.5% grade, as previously described [17, 28].

## Statistical analysis

Statistical analysis employed the IBM SPSS Statistics 23 (SPSS Software, Inc., USA, 2015) and Prism® 8 for Windows software (GraphPad Software, Inc., USA, 2019). The observed power (OP) was calculated by post hoc power analyses using G*Power 3.1.9.7 for Windows software [29].

The normality of the distribution of the variables was verified by the Shapiro-Wilk test, by visually Q-Q plot analysis, and scores greater than 1.5 times the interquartile range out of the boxplot were considered outliers [30, 31]. The homogeneity analysis of the covariance matrices of each dependent variable was performed by *Box's M test*, and the Levene test verified the analysis of variance homogeneity. We used mean and standard deviation as descriptive statistics to present compliance with normality assumptions. Otherwise, data are presented as median and quartiles (25% and 75%).

Depending on the data distribution, inferential analyses were run either with the independent t-test or the Mann-Whitney test. The dependent variables that presented normality assumptions, homoscedasticity, and no outlier were present, and the groups presented similar sizes, which protect the inflation of the type I error due to multiple testing of variables dependent; we used *the multivariate analysis of variance test (MANOVA)* in the comparative analysis before, during and, after the submaximal exercise test. The analysis of variance (ANOVA) of each set of the dependent variable was conducted by the *MANOVA test*.

A simple linear regression was also performed on data that met the assumptions of normality, linearity of parameters, normality of residuals, independent values, homoscedasticity, and absence of autocorrelation of residuals (Durbin-Watson test), absence of multicollinearity, and absence of outliers.

The effect size (ES) used for the independent t-test was Cohen's d, adopting the following parameters: $d < 0.2$: trivial effect; $d \geq 0.2$ and $<0.5$: small effect; $d \geq 0.5$ and $< 0.8$: medium effect and $d \geq 0.8$: large effect. ES used for the Mann-Whitney test was score Z and we adopted the following criteria for interpreting: $< 0.1$: trivial effect; $\geq 0.1 <0.3$: small effect; $\geq 0.3 < 0.5$: medium effect and $\geq 0.5$: large effect [32]. ES used for MANOVA test was multivariate square eta ($M\eta^2$) and we adopted the following criteria: $< 0.02$: trivial effect; $\geq 0.02 < 0.13$: small effect; $\geq 0.13 <0.26$: medium effect; $\geq 0.26$: large effect [33]. The two-tailed level of statistical significance was set at a $p \leq 0.05$.

## Results

Table 1 shows values of the age, body mass index, resting heart rate, heart rate peak, speed$_{peak,}$ and the total time of effort (EXT-time) for both the BG and NG. We observed a large ES and lower value of RHR on the BG compared to NG ($p \leq 0.01$). We did not observe any difference between groups' age, BMI, HR$_{peak}$, speed$_{peak,}$ and EXT-time data ($p \geq 0.51$).

Table 2 shows comparative values of the chronotropic response before, during, and after the submaximal treadmill exercise test for groups. The MANOVA for three dependents variables at rest (HR$_{ort}$, $\Delta_{abs}$RHR, and $\Delta_{\%}$RHR), for two dependents variables during submaximal treadmill exercise test (HR$_{initial}$ and CR), and three dependents variables for absolute (HHR$_1{}^{st}{}_{min}$, HHR$_3{}^{sd}{}_{min}$, and HHR$_5{}^{th}{}_{min}$) and relative (%HHR$_1{}^{st}{}_{min}$, %HHR$_3{}^{sd}{}_{min}$, and %HHR$_5{}^{th}{}_{min}$) chronotropic response following submaximal treadmill exercise test presented a significant difference between groups according to Pillai's trace ($p < 0.01$).

At rest, we observed a large ES in which BG presented lower values of HR$_{ort}$ and higher values $\Delta_{\%}$RHR ($p \leq 0.03$) compared to NG. No differences were observed between groups on the $\Delta_{abs}$RHR ($p = 0.31$). During the submaximal treadmill exercise test, we observed a large ES in

**Table 1. Comparative values of age, body mass index, resting heart rate, heart rate peak, and total effort time in the BG (n = 20) and NG (n = 20).**

| Variables | BG | NG | ES† | Difference between means (95% CI) | *p |
|---|---|---|---|---|---|
| Age (years) | 26.1 ± 5,6 | 27.3 ± 6.0 | 0.1 | - 1.1 (- 4.8 to 2.5) | 0.53 |
| BMI (kg/m²) | 24.1 ± 2.2 | 24.5 ± 1.6 | 0.2 | -0.4 (- 1.6 to 0.8) | 0.51 |
| RHR (bpm) | 56.1 ± 3.2 | 66.9 ± 4.9 | 2.6 | - 10.8 (- 13.4 to -8.13) | ≤ 0.01 |
| HR$_{peak}$ (bpm) | 164.1 ± 5.1 | 164.2 ± 4.8 | 0.01 | - 0.1 (- 3.2 to 3.1) | 0.95 |
| EXT-time (s) | 455.1 ± 54.5 | 443.9 ± 52.8 | 0.1 | 11.2 (- 23.1 to 45.6) | 0.51 |
| Speed$_{peak}$ (km/h) | 11.2 ± 0.9 | 11.0 ± 0.9 | 0.2 | 0.2 (- 0.3 to 0.8) | 0.40 |

BG: bradycardia group; NG: normal heart rate group; ES: effect size; CI: confidence interval; BMI: body mass index; kg: kilograms; m: meters; HR: heart rate; bpm: beats per minute; EXT-time: time effort total; s: seconds; km: kilometers; h: hours; TE: trivial effect; LE: large effect

†: Cohen's d

* *independent test t* ($p \leq 0.05$).

**Table 2. Mean (± standard deviation) comparative values of the chronotropic response before, during, and after a submaximal exercise test.**

| Variable | BG | NG | ES† | OP | Difference between means (95% CI) | *p |
|---|---|---|---|---|---|---|
| **Rest** | | | | | | |
| $HR_{ort}$ (bpm) | 76.7 ± 8.5 | 84.9 ± 9.8 | 0.29 | 99% | - 8.2 (- 14.1 to—2.3) | < 0.01 |
| $\Delta_{abs}RHR$ (bpm) | 20.6 ± 8.1 | 18.0 ± 8.0 | | | 2.6 (-2.5 to 7.7) | 0.31 |
| $\Delta_{\%}RHR$ (%) | 37.0 ± 15.4 | 27.6 ± 12.1 | | | 5.5 (0.5 to 10.4) | 0.03 |
| **Exercise** | | | | | | |
| $HR_{initial}$ (bpm) | 82.0 ± 5.5 | 88.0 ± 6.3 | 0.29 | 99% | - 6.2 (- 10.1 to—2.4) | < 0.01 |
| CR (bpm) | 108.0 ± 5.5 | 97.0 ± 5.7 | | | 10.7 (7.1 to 14.3) | < 0.01 |
| **Recovery** | | | | | | |
| $HRR_1{}^{st}{}_{min}$ (bpm) | 36.5 ± 7.8 | 35.5 ± 8.1 | 0.10 | 89% | 1.0 (-4.0 to 6.1) | 0.68 |
| $HRR_3{}^{rd}{}_{min}$ (bpm) | 62.4 ± 6.9 | 57.4 ± 7.1 | | | 4.9 (0.4 to 9.4) | 0.03 |
| $HRR_5{}^{th}{}_{min}$ (bpm) | 66.9 ± 5.6 | 60.4 ± 5.8 | | | 6.4 (2.7 to 10.1) | < 0.01 |
| $\%HRR_1{}^{st}{}_{min}$ (bpm) | 22.3 ± 4.8 | 21.6 ± 4.9 | 0.11 | 93% | 0.6 (-2.4 to 3.8) | 0.66 |
| $\%HRR_3{}^{rd}{}_{min}$ (bpm) | 38.0 ± 4.4 | 34.9 ± 4.1 | | | 3.0 (0.3 to 5.8) | 0.02 |
| $\%HRR_5{}^{th}{}_{min}$ (bpm) | 40.7 ± 3.2 | 36.8 ± 3.4 | | | 3.9 (1.8 to 6.0) | < 0.01 |

BG: bradycardia group; NG: normal heart rate group; ES: effect size; OP: observed power; CI: confidence interval; HR: heart rate; bpm: beats per minute; ort: orthostatic; $\Delta_{abs}RHR$: absolute variation resting heart rate; $\Delta_{\%}RHR$: relative variation resting heart rate; CR: chronotropic reserve; min: minute; HRR: heart rate recovery; %HRR: relative heart rate recovery; LE: large effect; SE: small effect

†: Multivariate square eta

* *MANOVA test* ($p \leq 0.05$).

which BG presented lower values of $HR_{initial}$ and higher values CR than NG ($p \leq 0.01$). Following the submaximal treadmill exercise test, we observed a small ES, and we did not observe differences between groups on the $HHR_1{}^{st}{}_{min}$ ($p = 0.68$) and $\%HHR_1{}^{st}{}_{min}$ ($p = 0.66$), but we observed a higher value of $HHR_3{}^{sd}{}_{min}$ and $HHR_5{}^{th}{}_{min}$ and $\%HHR_3{}^{sd}{}_{min}$ and $\%HHR_5{}^{th}{}_{min}$ on BG compared to NG ($p \leq 0.02$), as shown in Table 2.

Table 3 shows comparative values of the parasympathetic cardiac activity before, during, and after the submaximal treadmill exercise test, both on the BG and NG. At rest, BG presented higher $rMSSD_{sup}$, $rMSSD_{ort}$, and $\Delta_{abs}rMSSD_{ort-sup}$ compared to NG ($p < 0.01$) with an ES of moderate to large. We did not observe differences between groups on the $\Delta_{\%}rMSSD_{ort-sup}$ ($p = 0.22$). During the submaximal treadmill exercise test, we did not observe differences between groups and on the $rMSSD_{peak}$ ($p = 0.14$), but we observed higher values of $\Delta_{abs}rMSSD_{peak-sup}$ and $\Delta_{\%}rMSSD_{peak-sup}$ ($p < 0.01$) on BG than NG and a large and moderate ES, respectively. Following the submaximal treadmill exercise test, we did not observe differences between groups on the $rMSSD_1{}^{st}{}_{min}$ ($p = 0.11$). BG presented higher $rMSSD$ $3^{rd}min$ and $rMSSD$ $5^{th}min$ values than NG ($p < 0.01$) and a moderate ES in both indexes.

Table 4 shows the simple linear regression derived from the values of the RHR group (predictor variable) with the chronotropic response (outcome variable) before, during, and after the submaximal treadmill exercise test. The results demonstrated that the model significantly improves our ability to explain the behavior of the outcome variables ($p \leq 0.001$). Thus, the behavior of the outcome variable can be explained by the predictor variable.

The NG presents on average 8.2 bpm in $HR_{ort}$ and 6.2 bpm in $HR_{initial}$ more than the BG, in which the RHR explains 17% and 22%, respectively, of the variance of this difference average between the groups. NG presents on average 5.5% in $\Delta_{\%}RHR$, 10.7 bpm in CR, 4.9 bpm in $HRR$ $3^{sd}{}_{min}$, 6.4 bpm in $HRR$ $5^{th}{}_{min}$, 3.1 bpm in $\%HRR3^{sd}{}_{min}$, and 3.9 bpm in $\%HRR5^{th}{}_{min}$ lower than the BG, in which RHR explain from 11% to 48% o the variance of this difference average between the groups.

**Table 3. Median (25th– 75th) comparative values of the parasympathetic cardiac activity before, during, and after a submaximal exercise test.**

| Variables | BG | NG | ES† | OP | Difference between median (95% CI) | *p |
|---|---|---|---|---|---|---|
| **Rest** | | | | | | |
| rMSSD$_{sup}$(ms) | 68.7 (34.7–187.9) | 38.5 (18.5–113.9) | 0.5 | 96% | 30.1 (12.7 to 48.2) | < 0.01 |
| rMSSD$_{ort}$(ms) | 25.5 (10.2–105.0) | 15.6 (9.7–57.8) | 0.3 | 68% | 9.8 (0.8 to 17.1) | 0.01 |
| ΔrMSSD$_{res}$(ms) | 42.3 (6.4–130.0) | 20.6 (-0.9–90.4) | 0.4 | 77% | 21.8 (4.2 to 38.2) | < 0.01 |
| Δ%rMSSD$_{res}$(%) | 60.3 (18.4–88.0) | 58.3 (-4.8–79.4) | 0.1 | 11% | 1.9 (-8.9 to 18.0) | 0.43 |
| **Exercise** | | | | | | |
| rMSSD$_{peak}$(ms) | 3.4 (2.2–5.7) | 3.1 (1.9–4.3) | 0.1 | 18% | 0.2 (-0.2 to 0.8) | 0.27 |
| ΔrMSSD$_{exer}$(ms) | 65.6 (30.5–185.0) | 36.0 (15.0–111.0) | 0.5 | 95% | 29.6 (12.4 to 48.1) | < 0.01 |
| Δ%rMSSD$_{exer}$(%) | 95.3 (87.7–98.5) | 92.2 (81.1–97.4) | 0.3 | 63% | 3.1 (0.2 to 5.3) | 0.02 |
| **Recovery** | | | | | | |
| rMSSD$_1$$^{st}$$_{min}$(ms) | 3.9 (2.8–9.9) | 3.5 (2.0–9.6) | 0.1 | 20% | 0.4 (-0.3 to 0.9) | 0.23 |
| rMSSD$_3$$^{rd}$$_{min}$(ms) | 9.6 (5.2–21.0) | 6.9 (3.0–25.0) | 0.4 | 73% | 2.7 (0.6 to 4.4) | 0.01 |
| rMSSD$_5$$^{th}$$_{min}$(ms) | 8.7 (5.3–20.0) | 6.2 (3.1–23.0) | 0.4 | 86% | 2.6 (1.1 to 4.9) | < 0.01 |

BG: bradycardia group; NG: normal heart rate group; ES: effect size; OP: observed power; CI: confidence interval; rMSSD: the square root of the mean of the square of successive adjacent R-R intervals difference; sup: supine position; ms: millisecond; ort: orthostatic position; ΔrMSSD$_{res}$: absolute variation of resting rMSSD; Δ% rMSSD$_{res}$: relative variation of resting rMSSD; ΔrMSSD$_{exer}$: absolute variation of exercise rMSSD; Δ%rMSSD$_{exer}$: relative variation of exercise rMSSD; min: minute; LE: large effect; ME: medium effect; TE: trivial effect

†: Score Z

* *Mann-Whitney test* (p ≤ 0.05).

## Discussion

Our study observed new and relevant findings regarding different RHR average values on the HR dynamics and parasympathetic activity at rest, during, and after a submaximal exercise test in young physically active men.

We observed that BG (RHR average values < 60 bpm) showed low HR and high parasympathetic activity at rest (supine and orthostatic positions); High chronotropic and parasympathetic responses (reduction) after changing posture from supine to orthostatic at rest; During the submaximal exercise test was observed higher chronotropic reserve, and parasympathetic

**Table 4. Simple linear regression analysis derived from differences between means of the groups on a chronotropic response before, during, and after a submaximal exercise test.**

| Predictor variable | Dependents variable | Slope | R² | OP | 95% confidence interval of the mean difference to BG | *p |
|---|---|---|---|---|---|---|
| **BG vs. NG** | HR$_{ort}$ | 8.2 | 17% | 99% | 2.1 to 14.1 | ≤ 0.01 |
| | Δ$_{\%}$RHR (%) | -5.5 | 11% | 99% | -10.4 to -0.5 | 0.03 |
| | HR$_{initial}$ (bpm) | 6.2 | 22% | 99% | 2.4 to 10.0 | ≤ 0.01 |
| | CR (bpm) | -10.7 | 48% | 99% | -14.3 to -7.1 | ≤ 0.01 |
| | HRR$_3$$^{sd}$$_{min}$ (bpm) | -4.9 | 11% | 99% | -9.4 to -0.4 | 0.03 |
| | HRR$_5$$^{th}$$_{min}$ (bpm) | -6.4 | 24% | 99% | -10.1 to -2.7 | ≤ 0.01 |
| | %HRR$_3$$^{rd}$$_{min}$ (bpm) | -3.1 | 12% | 99% | -5.8 to -0.3 | 0.02 |
| | %HRR$_5$$^{th}$$_{min}$ (bpm) | -3.9 | 27% | 99% | -6.1 to -1.8 | ≤ 0.01 |

Slope: regression coefficient; R²: determination coefficient; OP: observed power; NG: normal heart rate group; BG: bradycardia group; HR: heart rate; ort: orthostatic position; bpm: beats per minute; Δ$_{\%}$RHR: relative variation of resting heart rate; CR: chronotropic reserve; HRR: heart rate recovery; min: minute; %HRR: relative heart rate recovery

* *Linear regression test* (p ≤ 0.05).

withdrawal, and following the exercise, faster HRR and parasympathetic reactivation were observed compared to NG (RHR average values $\geq$ 60 bpm).

The functional basis of the interaction between low RHR average values with chronotropic and parasympathetic activity responses is difficult to explain and can only be conjectured, considering the complexity of the mechanism involved in HR dynamics. Hence, our results suggest that RHR average values < 60 bpm in the supine position (bradycardia) worked as a marker for greater cardiac autonomic modulation in distinct functional conditions (rest, exercise, and recovery).

It is well-established that the HR increases (chronotropic reserve) during an exercise test due to parasympathetic withdrawal and sympathetic activation, and following the exercise test, the short-term HR adjustment responds to the rapid parasympathetic recovery and progressive sympathetic withdrawal [5]. Therefore, the coactivation of both autonomic branches, parasympathetic and sympathetic, occurs over several minutes immediately after exercise [5, 34].

Previous studies with similar exercise protocols to our study verified a positive significant correlation between resting parasympathetic activity with the HRR and parasympathetic activity after the submaximal exercise test [9, 11]. Evrengul et al. [11] observed a positive significant correlation between parasympathetic activity at rest and HRR in the 3$^{rd}$ minute during passive recovery (supine position). Danieli et al. [9] observed a positive correlation between resting parasympathetic activity with HRR at 30s, 1$^{st}$ and 2$^{nd}$ minute during passive recovery (seating position).

On the other hand, some authors have shown divergent results considering the relationship between resting parasympathetic activity with HRR and parasympathetic activity after the exercise test [8–16]. One of the pioneers in the field was Javorka et al. [12], who did not observe a significant correlation between resting parasympathetic activity with relative HRR and parasympathetic reactivation in the 1$^{st}$ and 5$^{th}$minute after submaximal exercise. In the same direction, Molina et al. [14, 15] published studies that did not observe a correlation between absolute and relative HRR at the 1$^{st}$ minute with parasympathetic activity in the supine position and the standing position following maximal treadmill exercise testing in healthy men.

Considering the present results, we did not observe differences between groups on the HHR, %HRR, and parasympathetic activity in the first minute of recovery; it is an established standard time. Probably the absence of a significant difference in the parasympathetic activity within the first minute following the submaximal exercise test may be due to the coactivation of both autonomic branches (rapid parasympathetic reactivation with simultaneously sympathetic deactivation) and the effort interruption criterion in 85% of predicted maximum HR, which could impact this comparative analysis (BG vs NG).

On the other hand, most studies on HRR focused on the recovery in the first minute and the third minute after exercise [8–13] and there are a few literature reports in which both early and late HRR has been examined [14–16]. Additionally, different implications of exercise physiology can be appreciated with a comprehensive analysis of HRR beyond the first minute. After exercise, HRR in the first 30s to 1$^{st}$ min is predominantly driven by parasympathetic reactivation (early phase with rapid decline), whereas in the subsequent two or more minutes (late phase with slow decay), sympathetic withdrawal in addition to parasympathetic reactivation, sets in as a pivotal determinant of heart rate decline [5]. Therefore, BG compared to NG showed faster absolute and relative HRR in the 3$^{rd}$ and 5$^{th}$ minutes of recovery due to higher parasympathetic activity values in the 3$^{rd}$ and 5$^{th}$ minutes of recovery, as shown in Table 3.

In terms of cardiac autonomic analysis, assessing the RHR could be practical, straightforward, and feasible considering the comprehensive understanding of the effect of RHR in

different functional conditions, as shown in the present study. The monitoring of the RHR offers a relatively low-cost way to screen for risk factors with unfavorable prognoses such as increased cardiovascular morbimortality and sudden death [2, 7]. Hence, the novelty of the present study is that this results may open a new possibility to monitor the training status [4, 35] and predict cardiorespiratory fitness [36, 37].

Thus, one possibility is that the lower RHR average values (higher parasympathetic activity) and greater parasympathetic reactivity (reduction) after the postural change (orthostatic stress) before an adaptative functional demand as the exercise may be considered a preliminary marker of better chronotropic and parasympathetic responses during and after exercise.

Another aspect to be highlighted in our work is that BG showed higher chronotropic reserve and parasympathetic withdrawal during the submaximal incremental exercise test and faster HRR and parasympathetic reactivation after post-exercise compared to NG. Therefore, our results may add to the literature that RHR average values might be considered a cardiac autonomic "flexibility" index, which helps to understand the relationship between RHR, its adjustment, and cardiovascular health, which is important and potentially toll on individuals' care. From a clinical perspective, the results may be considered important to the individual evaluation of cardiovascular health, but this goes beyond our scope and should be investigated in future studies [38, 39].

In addition, our findings may add important information based on a novel approach toward developing an explication regression based on the effect of RHR average values on the HR response during these functional conditions (rest, exercise, and recovery) for a better understanding of these complex interactions.

Our study advances from previous studies showing that RHR average values explained the variance of HR at rest, exercise, and recovery from 11 to 48% in physically active men. We believe that the present study could open a new possibility of applying this analysis model in other populations (impaired cardiac autonomic regulation of HR or sick sinus syndrome), considering its preliminary information about the cardiac autonomic function "flexibility" to different functional conditions.

In the present study, we observed that $HR_{ort}$ in the NG presents on average 8.2 beats/min more than the BG; during exercise, NG showed a reduced chronotropic reserve of 10.7 beats/min compared to BG; and following the exercise, the HRR at 3rd and 5th minutes were 4.9 beats/min and 6.4 beats/min lower in NG than BG, respectively. Studies have shown that subtle variations in the HR, like 10 beats/min increase in RHR, delayed chronotropic response during exercise, and slow HRR after the exercise test may reflect a decreased parasympathetic modulation and increased sympathetic activity (relative autonomic imbalance) [1–3, 6, 7]. Regarding cardiac autonomic imbalance, it is well-established in the literature the association between reduced cardiac autonomic modulation with unfavorable prognoses such as increased cardiovascular morbimortality and diagnostics of overtraining syndrome and impaired cardiorespiratory fitness [35, 36, 40–43].

Although the present study cannot estimate the data, our analysis over a linear regression test clearly shows the effect of RHR average values on HR dynamics and parasympathetic activity in all assessed functional conditions. From a practical perspective, this observation could be significant considering some clinical conditions associated with elevated cardiometabolic risks, such as subclinical cardiac ischemia or severe hypertension, may preclude the accomplishment of maximum exercise testing. Alternatively, the submaximal exercise test is widely used in clinical settings for functional evaluation. In this context, to the best of our knowledge, no study has verified the effect of RHR on HR and parasympathetic activity during rest, exercise, and recovery using submaximal exercise testing.

Another aspect to be highlighted in our work is considering how the exercise dose (i.e., intensity, duration, and modality) may influence the HR and parasympathetic activity responses [5]. Thus, both groups achieved the same $HR_{peak}$ in the same EXT-time and same intensity ($Speed_{peak}$). In other words, since we managed the intensity, duration, and modality of the submaximal exercise test, the observed differences between the groups were not influenced by the exercise protocol, reinforcing the influence of RHR average values on the group's outcomes during all functional conditions.

Limitations in our work include the exclusion of older subjects, athletes, and women of any age range, so the findings cannot be extrapolated to these groups of people. Also, these results cannot be extrapolated to men in general because our sample was relatively homogenous, and thus men with different clinical/functional conditions would present different responses than those observed in the present study. Additionally, our findings cannot be extrapolated for other workload and duration exercises or protocols like a cycle or arm ergometry, where the individuals sit during rest, exercise and recovery. Also, the participants performed an active recovery after exercise in the orthostatic position, which limits the comparison to similar conditions. On the other hand, to reduce the influence of those potential confounders, we decided to prioritize internal validity instead of external validity. In addition, all differences observed were statistically significant with a power greater than 80%, mitigating, therefore, the possibility of a type I error. So, although in the present study, the sample homogeneity may represent a limitation, on the other side, this methodological condition could be considered one strength reinforcing our findings due to its strong internal validity.

## Conclusion

We concluded that BG (RHR average values < 60 bpm) showed higher chronotropic and parasympathetic activity at rest; higher chronotropic reserve and parasympathetic withdrawal during the submaximal exercise test, and faster HRR and parasympathetic reactivation after effort compared to NG in young physically active men. Lastly, low RHR average values (< 60 bpm) showed to be a cardiac autonomic "flexibility" index for assessing individual cardiovascular health in young physically active men.

## Supporting information

**S1 Data. Availability statement.**
(CSV)

## Author Contributions

**Conceptualization:** Giliard Lago Garcia, Carlos Janssen Gomes da Cruz, Guilherme Eckhardt Molina.

**Data curation:** Giliard Lago Garcia, Guilherme Eckhardt Molina.

**Formal analysis:** Giliard Lago Garcia.

**Funding acquisition:** Giliard Lago Garcia.

**Methodology:** Giliard Lago Garcia, Luiz Guilherme Grossi Porto, Carlos Janssen Gomes da Cruz, Guilherme Eckhardt Molina.

**Project administration:** Giliard Lago Garcia.

**Resources:** Giliard Lago Garcia.

**Visualization:** Giliard Lago Garcia.

**Writing – original draft:** Giliard Lago Garcia.

**Writing – review & editing:** Giliard Lago Garcia, Luiz Guilherme Grossi Porto, Carlos Janssen Gomes da Cruz, Guilherme Eckhardt Molina.

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
