## [Decision Letter · Decision Letter 0]

20 Jul 2022

PONE-D-22-17046Can resting heart rate patterns explain the heart rate and parasympathetic responses during rest, exercise, and recovery?PLOS ONE

Dear Dr. Garcia,

Thank you for submitting your manuscript to PLOS ONE. After careful consideration, we feel that it has merit but does not fully meet PLOS ONE’s publication criteria as it currently stands. Therefore, we invite you to submit a revised version of the manuscript that addresses the points raised during the review process.

Specifically, rewriting to render cleared the  rational, aim and discussion of the study is mandatory, as well as some methodological clarifications as emphasised by the reviewers.

We look forward to receiving your revised manuscript.

Kind regards,

Laurent Mourot

Section Editor

PLOS ONE

Journal Requirements:

"Yes. This work was supported by Fundação De Apoio À Pesquisa Do Distrito Federal - FAPDF under Grant 11/2022"

4. Thank you for stating the following in the Funding Section of your manuscript: 

"This work was supported by Fundação De Apoio À Pesquisa Do Distrito Federal - FAPDF under Grant 11/2022."

"Yes. This work was supported by Fundação De Apoio À Pesquisa Do Distrito Federal - FAPDF under Grant 11/2022"

Reviewers' comments:

Reviewer's Responses to Questions

**Comments to the Author**

1. Is the manuscript technically sound, and do the data support the conclusions?

Reviewer #1: Partly

Reviewer #2: Partly

2. Has the statistical analysis been performed appropriately and rigorously? 

Reviewer #1: Yes

Reviewer #2: No

3. Have the authors made all data underlying the findings in their manuscript fully available?

Reviewer #1: Yes

Reviewer #2: Yes

4. Is the manuscript presented in an intelligible fashion and written in standard English?

Reviewer #1: No

Reviewer #2: Yes

5. Review Comments to the Author

Reviewer #1: In the present investigation the authors sought to investigate the relationship between resting, exercise and recovery HR and HRV (only the time-domain HRV index RMSSD). They divided the participants into two groups according to their resting HR (Bradycardic vs Normocardic Group). The manuscript needs extensive revision before being considered for publication as some statements and considerations are not totally convincing. The manuscript also needs revision for language and grammar. Please consider the following suggestions.

Abstract:

L14: you may prefer using dynamics instead of dynamic

L14: please change into “remains to be clarified”

L17: what do you mean with patterns? That sounds like just the average value of HR at rest in the supine position..am I correct?

L23: please mention that this “manoeuvre” (i.e. the change in posture) is part of your evaluation protocol. Indeed, it seems that you wonder whether individuals with a low resting HR in the supine position have lower values of HR in the orthostatic position and/or different responses to the change in posture than individuals with a higher resting HR

Introduction:

L40: please clarify “chronotropic reserve”, that is usually defined considering both resting HR and HRmax

L43: and what happens to HR? this seems a good spot to introduce HRR

L44: what do you mean? A higher parasympathetic activity at rest would reflect into a faster parasympathetic recovery after exercise and/or to a faster para response to an active manoeuvres, such as an orthostatic challenge? I would delete this sentence..at this point of the introduction is still not clear how this two phenomena would be related

L47: consider diving this long sentence into shorter sentences

L54: you probably forgot “activity”; in my opinion HRV could be introduced earlier in the manuscript as the assessment of parasympathetic activity in the study you mention mainly relies on that

L58: and what did they found? what kind of HR (sub-max or max)? It is not clear how resting HR is related to exercising HR. Is it that individuals with a low HR tend to have low values of HR during exercise for a given sub-max exercise intensity?

L63: please provide some references to your statements

L74: what are these RHR profiles? You are using RHR as an abbreviation for resting heart rate, but here you are probably intending that is possible to characterize/classify an individual’s cardiac autonomic profile via measurements of HR and HRV indices. If I am not wrong, you want to test the hypothesis that resting, exercise and recovery HR and HRV responses can be somewhat related and that information obtained from HR and HRV evaluations at rest can inform about an individual’s ability to responds to stressful tasks such as exercise. Please consider rephrasing these last sentences.

L81: what is your hypothesis?

L83: please consider not using patterns if RHR is just the average value of HR (bpm) without any other information about its variability

Methods

L103: you might not need “RHR” here

L128: which threshold did you use?

L137: what is the physiological relevance of HR initial?

L137: maximum “predicted” HR.

L151: over a 5-min time window?

L155: over a 1-min time window? From end EX– 1 min to end EX ?

L158: please move this paragraph before the description of the analysis you conducted on HR/HRV indices, otherwise it is hard to follow

L158: this is a very important part for the study outcomes being HRR/HRV recovery influenced by the exercise intensity domain. Please clarify that 85% is the target to be reached in terms of predicted HRmax. Is there any other measurement that can inform about exercise intensity from a metabolic point of view? VO2?

L164: were participants allowed to run? The average duration of the test is 7.5 min, according to the protocol at 7.5 min the exercise intensity is 9 km/h. Was the 2 initial min enough to reach a steady value of HR before further increasing the treadmill velocity?

L195: trivial effect instead of no effect

Results

Please state in the tables whether the values refer to mean or median values

Discussion

L253: still not clear what RHR profiles means

L257: how? The two groups have the same average age..so the predicted HRmax is on average the same, the difference in resting HR directly reflects into a different “estimated” chronotropic reserve

What are the potential issues you may face targeting 85% of the predicted HR max in two groups that differ in their resting HR? are there any? Consider discussing this in the limitations section

L257: how did you measure para withdrawal during exercise? That is not clear at all. Is there a difference in HR kinetics? Is there a faster HR response in BG? The difference in reactivity refers to the orthostatic task.

L258: HRR 1st min should reflect parasympathetic reactivation but this is not different in the two groups. Please elaborate on this. HRR 5th min should reflect para reactivation + sympathetic withdrawal occurring during the post-exercise period, but you are not properly discussing this in the manuscript. HRR30s is also often used as a marker of parasympathetic reactivation. Why did you choose not to present those data?

L265: this part is really similar to the first part of the introduction. Please consider rephrasing

L271-L283: this part is not very comprehensible. Please try to make your point more clear rearranging these three sentences and avoiding repetitions

L288: for a similar resting HR can HRV analysis add some meaningful information to interpret subject’s cardiac autonomic profile and responses to stressful tasks?

L351: so why have you found the differences 3 and 5 min post exercise? How does resting HR can influence your calculations? If you try to consider resting HR do these differences disappear?

BG starts from 82 bpm and NG from 88 bpm, they both go up to 164 then down to 97 bpm and 104 bpm 5 min post-exercise. So after 5 min both groups are around + 15 bpm over their initial HR, and the difference seems to be similar to the difference at rest. If you consider the higher starting point for NG group the difference becomes further smaller. So, are they different? Or not?

This part is really important for the general purpose of the manuscript. It needs to be better presented and discussed.

What about adding some graphs?

How much of the variance in post-exercise HRR (1st, 3rd, 5th min of rec) or in reactivity to an orthostatic test can be explained by the differences in term of resting HR? what percentage of explained variance is enough to safely say that measuring HR/HRV at rest is helpful instead of including other type of measurements? Please clarify.

Reviewer #2: Dear Author,

Thank you for submitting your work. I have a few concerns regarding your work here:

1. The second objective of this study is not clear here; to determine whether RHR could explain the outcome of HR responses during 3 tested conditions. Please rewrite this objective so that it can be explained by your tabulated results.

2. What is your primary and secondary outcomes for this study? It seems that you are putting/testing everything together in this one study with only a small sample size. Make sure that you only focus on your main outcome measures.

3. What is the reason for choosing male subjects only? Why do you exclude female subjects?

4. How do you calculate your sample size?

5.Your tabulated results are not clear to the readers (see tables 2 and 3). The information in the table should be divided into the 3 conditions; at rest, during, and after.

6. For you regression analysis, how do you decide on the variables selection for analysis during this 3 conditions? Why did you have a mixture of absolute variable and relative variable (%) in your analysis?

7. You should be cautious in drawing conclusions because this study only includes young active males. Do not generalise your conclusion with your statement about using the autonomic flexibility index to assess individual CVS health.

6. PLOS authors have the option to publish the peer review history of their article (what does this mean?). If published, this will include your full peer review and any attached files.

Reviewer #1: No

Reviewer #2: No

---

## [Author Response · Author response to Decision Letter 0]

26 Aug 2022

Responses to the reviewers 

2022/08/26

Manuscript title: Can resting heart rate explain the heart rate and parasympathetic responses during rest, exercise, and recovery?

Manuscript ID: PONE-D-22-17046.

Authors: Giliard Lago Garcia, Luiz Guilherme Grossi Porto, Carlos Janssen Gomes da Cruz and Guilherme Eckhardt Molina

We thank the Editor-in-Chief of PLOS ONE for the opportunity to resubmit our paper, and we are very grateful to the reviewers for their comments and suggestions, which importantly improved our manuscript.

As requested, we have highlighted edits to facilitate further review. We believe that the responses and additions satisfactorily address the reviewers' comments and suggestions and hope you will agree. The edits have clarified several points and have resulted in an improved manuscript, which we hope is now ready for final acceptance.

# To Journal Requirements:

We adjusted as requested.

We adjusted as requested.

"Yes. This work was supported by Fundação De Apoio À Pesquisa Do Distrito Federal - FAPDF under Grant 11/2022"

We adjusted as requested.

4. Thank you for stating the following in the Funding Section of your manuscript: 

"This work was supported by Fundação De Apoio À Pesquisa Do Distrito Federal - FAPDF under Grant 11/2022."

"Yes. This work was supported by Fundação De Apoio À Pesquisa Do Distrito Federal - FAPDF under Grant 11/2022"

We adjusted as requested.

Upon re-submitting your revised manuscript, please upload your study's minimal underlying data set as either Supporting Information files or to a stable, public repository and include the relevant URLs, DOIs, or accession numbers within your revised cover letter. For a list of acceptable repositories, please see http://journals.plos.org/plosone/s/data-availability#loc-recommended-repositories. Any potentially identifying patient information must be fully anonymized.

We adjusted as requested.

# To the Reviewer 1

Abstract:

L14: you may prefer using dynamics instead of dynamic

We adjusted as requested.

# New word in the Abstract, page 2, line 25 (highlighted)

… dynamics…

L14: please change into "remains to be clarified"

# New word in the Abstract, page 2, line 26 (highlighted)

… remain to be clarified …

L17: what do you mean with patterns? That sounds like just the average value of HR at rest in the supine position.am I correct?

Thank you very much for the question. You are correct in your understanding, which was adjusted in the abstract.

# New words in the Abstract, page 2, line 28 (highlighted) 

… RHR average values …

L23: please mention that this "manoeuvre" (i.e. the change in posture) is part of your evaluation protocol. Indeed, it seems that you wonder whether individuals with a low resting HR in the supine position have lower values of HR in the orthostatic position and/or different responses to the change in posture than individuals with a higher resting HR

We adjusted as requested.

# New phase in the Abstract, page 2, lines 31 - 34 (highlighted)

A Polar RS800® was used to record the RR-interval series and HR at rest in the supine position, following the postural change (from supine to orthostatic position) and in the orthostatic position for 5 minutes, as well as during and after a submaximal exercise testing.

Introduction:

L40: please clarify "chronotropic reserve", that is usually defined considering both resting HR and HRmax

Thank you very much for the question.

# Removed words in the Introduction, pages 2, line 55 

… (chronotropic reserve).

# New phrase in the Introduction, pages 3, lines 69 – 72 (highlighted)

When evaluating the HR during the exercise, only one study showed a significantly positive association between parasympathetic activity at rest with chronotropic reserve (CR); the difference between HR at peak exercise and the RHR in the supine position, during the maximal exercise test [16].

In the section Material and Methods on page 07 and lines166 - 167, we defined chronotropic reserve as the difference between HRpeak from RHR, according to Jouven 2005 and Lauer, 1996.

JOUVEN, Xavier, et al. "Heart-rate profile during exercise as a predictor of sudden death." New England journal of medicine 352.19 (2005): 1951-1958.

LAUER, Michael S., et al. "Impaired heart rate response to graded exercise: prognostic implications of chronotropic incompetence in the Framingham Heart Study." Circulation 93.8 (1996): 1520-1526.

L43: and what happens to HR? this seems a good spot to introduce HRR

Thank you very much for their comments. Consequently, we have adjusted as suggested.

# New phase in the Introduction, page 2, lines 56 - 58 (highlighted)

In contrast, after exercise, the short-term post-exercise HR adaptation, the heart rate recovery (HRR), occurs in response to simultaneous rapid parasympathetic reactivation and progressive sympathetic deactivation [5].

L44: what do you mean? A higher parasympathetic activity at rest would reflect into a faster parasympathetic recovery after exercise and/or to a faster para response to an active manoeuvres, such as an orthostatic challenge? I would delete this sentence..at this point of the introduction is still not clear how this two phenomena would be related

We adjusted as requested by the Reviewer. 

# Removed phase in the Introduction, pages 3, line 60 

… Therefore, it is reasonable to expect that these chronotropic and autonomic responses to exercise-induced stress are dependent on the resting cardiac autonomic condition.

L47: consider diving this long sentence into shorter sentences

We adjusted as requested, and we rewrote this paragraph.

# New paragraph in the Introduction, page 3, lines 59 - 61 (highlighted)

Consequently, a high RHR, slow HR response during an exercise test, and a delayed HRR are associated with reduced parasympathetic activity, withdrawal impairment, and reactivation of parasympathetic activity, respectively [1-3, 6, 7].

L54: you probably forgot "activity"; in my opinion HRV could be introduced earlier in the manuscript as the assessment of parasympathetic activity in the study you mention mainly relies on that

We adjusted as requested. We did not introduce earlier HRV in the manuscript because we presented the cardiac autonomic modulation phenomenon that regulates heart rate at rest, during exercise, and during recovery in the previous paragraph.

# New word in the Introduction, pages 3, line 64 

… parasympathetic activity at rest...

L58: and what did they found? what kind of HR (sub-max or max)? It is not clear how resting HR is related to exercising HR. Is it that individuals with a low HR tend to have low values of HR during exercise for a given sub-max exercise intensity?

We adjusted as requested, and we rewrote the paragraph.

# New paragraph in the Introduction, pages 3, lines 67 – 75 (highlighted)

In this clinical setting, some studies have reported a significant association between resting parasympathetic activity with HRR after a maximal or submaximal incremental exercise test [9, 11, 14-16]. When evaluating the HR during the exercise, only one study showed a significantly positive association between parasympathetic activity at rest with chronotropic reserve (CR); the difference between HR at peak exercise and the RHR in the supine position, during the maximal exercise test [16]. On the other hand, other studies have not shown a significant association between parasympathetic activity at rest with HRR or parasympathetic activity after incremental exercise tests [8, 10, 12, 13].

L63: please provide some references to your statements

We adjusted as requested

# Two citations were inserted in the Introduction, page 3, line 80 (highlighted)

[5] Michael, Scott; GRAHAM, Kenneth S.; DAVIS, Glen M. Cardiac autonomic responses during exercise and post-exercise recovery using heart rate variability and systolic time intervals—a review. Frontiers in physiology, v. 8, p. 301, 2017.

[17] Garcia, Giliard Lago et al. Effect of different recovery protocols on the cardiac autonomic function. Revista Brasileira de Medicina do Esporte, v. 23, p. 16-20, 2017.

L74: what are these RHR profiles? You are using RHR as an abbreviation for resting heart rate, but here you are probably intending that is possible to characterize/classify an individual's cardiac autonomic profile via measurements of HR and HRV indices. If I am not wrong, you want to test the hypothesis that resting, exercise and recovery HR and HRV responses can be somewhat related and that information obtained from HR and HRV evaluations at rest can inform about an individual's ability to responds to stressful tasks such as exercise. Please consider rephrasing these last sentences.

Thank you very much for the question, and we agree with your statement. The text was adjusted. 

# New phrase in the Introduction, pages 4, lines 85, 87, 89 and 99 

… RHR average values …

# New phrase in the Introduction, pages 4, line 90

… preliminary tool for decision-making (i.e., stress management)…

L81: what is your hypothesis?

# New phrase in the Introduction, pages 4, lines 94 - 96

Therefore, we hypothesize that physically active young men with different RHR average values in the supine position show different HR and parasympathetic activity responses at rest, during, and after a submaximal exercise test.

L83: please consider not using patterns if RHR is just the average value of HR (bpm) without any other information about its variability

We adjusted as requested.

Methods

L103: you might not need "RHR" here

We adjusted as requested.

L128: which threshold did you use?

We adjusted as requested.

# New phrase in the Material and Methods, page 6, lines 149 - 152

We used the medium threshold method that only removed the visually observed ectopic points, as long as the tracing did not lose the physiological pattern and the removal did not exceed 1% of the recording. [19, 20].

L137: what is the physiological relevance of HR initial?

We thank the Reviewer's question. The physiological relevance of HR initial is related to the central neural control, termed central command, which operates in a feedforward manner changing heart rate response in distinctive degrees at the beginning of the exercise. Therefore, we included this variable in our study protocol to consider its relevance (neural response in the heart) prior to the exercise.

L137: maximum "predicted" HR.

We adjusted as requested.

# New word in the Material and Methods, page 7, line 167

… predicted …

L151: over a 5-min time window?

Yes. We recorded over a five minutes time widow, but we analyzed one-minute time widow during recovery (1st, 3rd, and 5th).

L155: over a 1-min time window? From end EX– 1 min to end EX ?

No. We adopted the 30 seconds time window when the volunteers reached 85% of their maximum predicted heart rate. 

# New words in the Material and Methods, page 7, lines 157 and 158

…,i.e., ≤ one minute,… … (30 seconds final)…

L158: please move this paragraph before the description of the analysis you conducted on HR/HRV indices, otherwise it is hard to follow

We adjusted as requested.

L158: this is a very important part for the study outcomes being HRR/HRV recovery influenced by the exercise intensity domain. Please clarify that 85% is the target to be reached in terms of predicted HRmax. Is there any other measurement that can inform about exercise intensity from a metabolic point of view? VO2?

Thank you for the questions. Regarding 85% being the target to be reached over the incremental sub-maximal test, we highlighted on page 7 lines 166 and 167 "and stopped when participants reached 85% of their maximum predicted HR (HRpeak) by the Tanaka formula [23]." Concerning other measurements from a metabolic point of view as the VO2, unfortunately, we did not measure the maximum oxygen consumption due to our laboratory's lack of a gas analyzer (ergospirometry). Thus, we did not perform the maximum test but a submaximal one (even though the studied population did not present risk factors) for our greater safety.

L164: were participants allowed to run? The average duration of the test is 7.5 min, according to the protocol at 7.5 min the exercise intensity is 9 km/h. Was the 2 initial min enough to reach a steady value of HR before further increasing the treadmill velocity?

This is an excellent question; the answer is yes; the volunteers were allowed to run when they felt like running. However, the volunteers usually started to run at around 7 km/h (3rd minute of the test) since the 2.5% grade remained constant throughout the test. 

We rewrite the paragraph.

# New paragraph in the Material and Methods, page 8, lines 186 - 190

The submaximal exercise test started with two minutes of warm-up at a speed of 3.0 km/h and 2.5% grade. The grade remained constant throughout the test and during recovery. After the two minutes of warm-up, the submaximal exercise test protocol started at a speed of 4.0 km/h and 2.5% grade, and the speed was increased by 1.0 km/h every minute until participants reached HRpeak.

# New data’s in the Results, Table 1. Page 10,

Speedpeak (km/h); BG: 11.2 ± 0.9;NG: 11.0 ± 0.9; ES: 0.2; 0.2 (- 0.3 to 0.8); p: 0.40

L195: trivial effect instead of no effect

We adjusted as requested.

# New words in the Material and Methods, pages 9 - 10, lines 220, 222 and 225

… trivial effect…

Results

Please state in the tables whether the values refer to mean or median values

Thank you very much for the question. We adjusted as requested

Discussion

L253: still not clear what RHR profiles means

We adjusted as requested

# New phrase in the Discussion, page 13, lines 276 and 277 

… RHR average values …

L257: how? The two groups have the same average age..so the predicted HRmax is on average the same, the difference in resting HR directly reflects into a different "estimated" chronotropic reserve. What are the potential issues you may face targeting 85% of the predicted HR max in two groups that differ in their resting HR? are there any? Consider discussing this in the limitations section

Thanks for the statement. Despite the same HRpeak, the NG showed a significantly reduced chronotropic reserve of 10.7 beats/min compared to BG; following the exercise, the HRR at 3rd and 5th minutes were 4.9 beats/min and 6.4 beats/min lower in NG than BG, respectively. 

Thus, the RHR average values can explain the variance of HR during exercise and recovery from 11 to 48% in physically active men, independent of the same HRpeak within groups. Therefore, the explication regression linear model applied in our study shows no potential issue considering targeting 85% of the predicted HRmax. Yet, we do not consider discussing the submaximal stress test as a limitation because we did not measure the maximum oxygen consumption due to our laboratory's lack of a gas analyzer (ergospirometry). Thus, we did not perform the maximum test but a submaximal one (even though the studied population did not present risk factors) for our greater safety.

L257: how did you measure para withdrawal during exercise? That is not clear at all. Is there a difference in HR kinetics? Is there a faster HR response in BG? The difference in reactivity refers to the orthostatic task.

It is an excellent question. We measured parasympathetic withdrawal by the difference between rMSSDpeak from rMSSDsup. According to Laborde, 2018: "Reactivity represents the change between baseline and a specific event, like completing a task, for example cognitive, emotional, or physical. Reactivity to an event or stress is crucial regarding adaptability, and both lower and higher vagal withdrawal can be facilitative when facing demands". Therefore, as well as the parasympathetic response to an orthostatic test and a submaximal exercise test represents a reactivity process.

Laborde, Sylvain, Emma Mosley, and Alina Mertgen. "Vagal tank theory: the three rs of cardiac vagal control functioning–resting, reactivity, and recovery." Frontiers in neuroscience 12 (2018): 458.

# New phrase in the Material and Methods, pages 7 and 8, lines 176 - 178

and we obtained the parasympathetic withdrawal by subtracting rMSSDpeak from rMSSDsup through the absolute (∆absrMSSDexer) and the relative (∆%rMSSDexer) rMSSD variation.

L258: HRR 1st min should reflect parasympathetic reactivation but this is not different in the two groups. Please elaborate on this. HRR 5th min should reflect para reactivation + sympathetic withdrawal occurring during the post-exercise period, but you are not properly discussing this in the manuscript. HRR30s is also often used as a marker of parasympathetic reactivation. Why did you choose not to present those data?

We are thankful to the Reviewer for this plausible and interesting question. We choose not to present HRR30s because there is no difference between groups. Also, the first minute following the exercise is established as a standard time for clinical evaluation; maybe the first minute of recovery is not the most appropriate post-exercise time for evaluating HRR as usually measured. 

We relocate and rewrite the paragraphs and add a new paragraph to the text.

# New paragraph relocated and rewritten in the Discussion, pages 14 and 15, lines 298 - 313

Previous studies with similar exercise protocols to our study verified a positive significant correlation between resting parasympathetic activity with the HRR and parasympathetic activity after the submaximal exercise test [9, 11]. Evrengul et al.[11] observed a positive significant correlation between parasympathetic activity at rest and HRR in the 3rd minute during passive recovery (supine position). Danieli et al. [9] observed a positive correlation between resting parasympathetic activity with HRR at 30s, 1st and 2nd minute during passive recovery (seating position).

On the other hand, some authors have shown divergent results considering the relationship between resting parasympathetic activity with HRR and parasympathetic activity after the exercise test [8-16]. One of the pioneers in the field was Javorka et al. [12], who did not observe a significant correlation between resting parasympathetic activity with relative HRR and parasympathetic reactivation in the 1st and 5thminute after submaximal exercise. In the same direction, Molina et al. [14, 15] published studies that did not observe a correlation between absolute and relative HRR at the 1st minute with parasympathetic activity in the supine position and the standing position following maximal treadmill exercise testing in healthy men.

# New paragraph in the Discussion, pages 15 and 16, lines 314 - 332

Considering the present results, we did not observe differences between groups on the HHR, %HRR, and parasympathetic activity in the first minute of recovery; it is an established standard time. Probably the absence of a significant difference in the parasympathetic activity within the first minute following the submaximal exercise test may be due to the coactivation of both autonomic branches (rapid parasympathetic reactivation with simultaneously sympathetic deactivation) and the effort interruption criterion in 85% of predicted maximum HR, which could impact this comparative analysis (BG vs NG).

On the other hand, most studies on HRR focused on the recovery in the first minute and the third minute after exercise [8-13] and there are a few literature reports in which both early and late HRR has been examined [14-16]. Additionally, different implications of exercise physiology can be appreciated with a comprehensive analysis of HRR beyond the first minute. After exercise, HRR in the first 30s to 1st min is predominantly driven by parasympathetic reactivation (early phase with rapid decline), whereas in the subsequent two or more minutes (late phase with slow decay), sympathetic withdrawal in addition to parasympathetic reactivation, sets in as a pivotal determinant of heart rate decline [5]. Therefore, BG compared to NG showed faster absolute and relative HRR in the 3rd and 5th minutes of recovery due to higher parasympathetic activity values in the 3rd and 5th minutes of recovery, as shown in Table 3.

L265: this part is really similar to the first part of the introduction. Please consider rephrasing

We adjusted as requested

# New phrase in the Introduction, page 2, lines 54 - 58

During an incremental exercise test, heart rate (HR) increases due to parasympathetic withdrawal and sympathetic activation [5]. In contrast, after exercise, the short-term post-exercise HR adaptation, the heart rate recovery (HRR), occurs in response to simultaneous rapid parasympathetic reactivation and progressive sympathetic deactivation [5].

# New paragraph in the Discussion, page 14, lines 292 - 297

It is well-established that the HR increases (chronotropic reserve) during an exercise test due to parasympathetic withdrawal and sympathetic activation, and following the exercise test, the short-term HR adjustment responds to the rapid parasympathetic recovery and progressive sympathetic withdrawal [5]. Therefore, the coactivation of both autonomic branches, parasympathetic and sympathetic, occurs over several minutes immediately after exercise [5, 34].

L271-L283: this part is not very comprehensible. Please try to make your point more clear rearranging these three sentences and avoiding repetitions

We adjusted as requested.

# New paragraph in the Discussion, pages 16 and 17, lines 345 - 357

Thus, one possibility is that the lower RHR average values (higher parasympathetic activity) and greater parasympathetic reactivity (reduction) after the postural change (orthostatic stress) before an adaptative functional demand as the exercise may be considered a preliminary marker of better chronotropic and parasympathetic responses during and after exercise. 

Another aspect to be highlighted in our work is that BG showed higher chronotropic reserve and parasympathetic withdrawal during the submaximal incremental exercise test and faster HRR and parasympathetic reactivation after post-exercise compared to NG. Therefore, our results may add to the literature that RHR average values might be considered a cardiac autonomic "flexibility" index, which helps to understand the relationship between RHR, its adjustment, and cardiovascular health, which is important and potentially toll on individuals' care. From a clinical perspective, the results may be considered important to the individual evaluation of cardiovascular health, but this goes beyond our scope and should be investigated in future studies [38, 39].

In addition, our findings may add important information based on a novel approach toward developing an explication regression based on the effect of RHR average values on the HR response during these functional conditions (rest, exercise, and recovery) for a better understanding of these complex interactions.

L288: for a similar resting HR can HRV analysis add some meaningful information to interpret subject's cardiac autonomic profile and responses to stressful tasks?

This is an excellent question. Yes, HRV can add meaningful information to subjects' cardiac autonomic profiles by considering that two individuals can show the same RHR with distinctive parasympathetic activity (i.e., Parasympathetic saturation).

L351: so why have you found the differences 3 and 5 min post exercise? How does resting HR can influence your calculations? If you try to consider resting HR do these differences disappear?

This is an excellent question. These differences in HRR between groups probably were due to higher parasympathetic activity values in the 3rd and 5th minutes of recovery. And the lower RHR average values (higher parasympathetic activity) and greater parasympathetic withdrawal (reactivity) after the postural change (orthostatic stress) at rest before an adaptative functional demand as the exercise, could mean marker of better chronotropic and parasympathetic responses after exercise.

# New phrase in the Discussion, page 16, lines 330 - 332

Therefore, BG compared to NG showed faster absolute and relative HRR in the 3rd and 5th minutes of recovery due to higher parasympathetic activity values in the 3rd and 5th minutes of recovery, as shown in Table 3.

BG starts from 82 bpm and NG from 88 bpm, they both go up to 164 then down to 97 bpm and 104 bpm 5 min post-exercise. So after 5 min both groups are around + 15 bpm over their initial HR, and the difference seems to be similar to the difference at rest. If you consider the higher starting point for NG group the difference becomes further smaller. So, are they different? Or not?

This part is significant for the general purpose of the manuscript, and it needs to be better presented and discussed.

We do not know. We did not do the correction by chronotropic reserve because possibly we may remove meaningful variance in outcomes of interest that can be attributed to autonomic and neurophysiological phenomena, according to Geus, 2019.

de Geus, Eco JC, et al. "Should heart rate variability be "corrected" for heart rate? Biological, quantitative, and interpretive considerations." Psychophysiology 56.2 (2019): e13287.

What about adding some graphs?

We are thankful to the Reviewer, but it was a style option for data presentation.

How much of the variance in post-exercise HRR (1st, 3rd, 5th min of rec) or in reactivity to an orthostatic test can be explained by the differences in term of resting HR? what percentage of explained variance is enough to safely say that measuring HR/HRV at rest is helpful instead of including other type of measurements? Please clarify.

This is an excellent question. Thank you! The RHR average values do not explain the variance of difference average between the groups the HHR at 1st minute of recovery. According to described in the article (page 17, lines 358 – 369): "The NG presents on average 8.2 bpm in HRort and 6.2 bpm in HRinitial more than the BG, in which the RHR explains 17% and 22%, respectively, of the variance of this difference average between the groups. NG presents on average 5.5 % in ∆%RHR, 10.7 bpm in CR, 4.9 bpm in HRR 3sdmin, 6.4 bpm in HRR 5thmin, 3.1 bpm in %HRR3sdmin, and 3.9 bpm in %HRR5thmin lower than the BG, in which RHR explain from 11% to 48% o the variance of this difference average between the groups".

Considering the present results of the percentage explained of HR, we can say that RHR average values open the possibility of a new approach (or analysis), which may add helpful information as a preliminary tool for decision-making in stress management for healthcare professionals bringing essential and complementary information related to individual cardiac autonomic capacity.

# To the Reviewer 2

1. The second objective of this study is not clear here; to determine whether RHR could explain the outcome of HR responses during 3 tested conditions. Please rewrite this objective so that it can be explained by your tabulated results.

We agree with the Reviewer's consideration of our statement. It improves our study. And, the second objective of this study was not clear, and therefore, we change it according Shmueli, 2010 which describe the explication regression.

# New phrase in the Introduction, page 4, lines 97-101 (highlighted)

Accordingly, our objectives were: (a) to compare the HR and parasympathetic responses during rest, exercise, and post submaximal exercise in physically active young men with different RHR average values; (b) to develop an explication regression based on the effect of RHR average values on the HR response during rest, exercise, and post submaximal exercise in physically active young men.

Shmueli, Galit. "To explain or to predict?." Statistical science 25.3 (2010): 289-310.

2. What is your primary and secondary outcomes for this study? It seems that you are putting/testing everything together in this one study with only a small sample size. Make sure that you only focus on your main outcome measures.

Thank you very much for the question. The primary outcome, we observed that BG (RHR < 60 bpm) showed high HR and parasympathetic response (modulation), higher chronotropic reserve, and parasympathetic withdrawal (reactivity) during the submaximal exercise test, and faster HRR and parasympathetic reactivation after the submaximal exercise test compared to NG. 

Furthermore, the second outcome RHR explained the variance of HR at rest, exercise, and recovery from 11 to 48% in physically active men.

Regarding the small sample size, this study shows that all differences observed were statistically significant with a power greater than 80%, mitigating the possibility of a type I error.

3. What is the reason for choosing male subjects only? Why do you exclude female subjects?

Thank you very much for the question. At the time of conducting this research, we only had this sample available. Second, according to Laborde, 2017, age, sex, physical fitness, and clinical status are considered important confounding variables influencing the analysis of HRV, and they must be controlled. 

LABORDE, Sylvain; MOSLEY, Emma; THAYER, Julian F. Heart rate variability and cardiac vagal tone in psychophysiological research–recommendations for experiment planning, data analysis, and data reporting. Frontiers in psychology, v. 8, p. 213, 2017.

4. How do you calculate your sample size?

Thank you very much for the question. As we wrote in the text (page 8; lines 198 – 199): "The observed power (OP) was calculated by post hoc power analyses using G*Power 3.1.9.7 for Windows software [29]". The post hoc power analyses often make sense after a study has already been conducted. In post hoc analyses, 1 – β is computed as a function of α, the population effect size parameter, and the sample size(s) used in a study. It thus becomes possible to assess whether a published statistical test had a fair chance of rejecting an incorrect H0. Importantly, post hoc analyses, like a priori analyses, require an H1 effect size specification for the underlying population. Post hoc power analyses should not be confused with so-called retrospective power analyses, in which the effect size is estimated from sample data and used to calculate the observed power, a sample estimate of the true power 

At any alpha level, increased sample sizes always yield greater statistical test power, and a potential problem then turns into excessive power. By "excessive," increasing the sample size implies that smaller and smaller effects will be perceived as statistically significant until, in very large samples, almost every effect is significant. The researcher must always be aware that the sample size can impact the statistical test, making it insensitive (with small samples) or overly sensitive (with very large samples).

The relationships between alpha, sample size, effect size, and power are very complicated, and many guidance references are available. Cohen examines power for most tests of statistical inference and guides acceptable levels of power, suggesting that studies should be designed to achieve alpha levels of at least 0.05 with power levels of 80%. To achieve such levels of power, the three factors – alpha, sample size, and effect size – must be considered simultaneously.

According to Laborde, 2017, in terms of sample sizes, the effect size distribution analysis suggests that to achieve 80% power, samples of 21 participants are required to detect large effect sizes. If another effect size or statistical power level is desired, power analysis can be conducted with the help of the G*Power software.

HAIR, Joseph F. Multivariate data analysis. 2009.

FIELD, Andy; MILES, Jeremy; FIELD, Zoë. Discovering statistics using R. 2012.

LABORDE, Sylvain; MOSLEY, Emma; THAYER, Julian F. Heart rate variability and cardiac vagal tone in psychophysiological research–recommendations for experiment planning, data analysis, and data reporting. Frontiers in psychology, v. 8, p. 213, 2017.

5.Your tabulated results are not clear to the readers (see tables 2 and 3). The information in the table should be divided into the 3 conditions; at rest, during, and after.

Thank you very much for the question. We adjusted as requested.

6. For you regression analysis, how do you decide on the variables selection for analysis during this 3 conditions? Why did you have a mixture of absolute variable and relative variable (%) in your analysis?

Thank you very much for the question. We utilized the theoric and only variables that presented significant statistics between groups because we have only one predictor variable (independent variable). Also, all variables utilized on simple linear regression met the assumptions of normality, linearity of parameters, normality of residuals, independent values, homoscedasticity, and absence of autocorrelation of residuals (Durbin-Watson test), absence of multicollinearity, and absence of outliers. Lastly, we used absolute and relative variables (%) data to reinforce the studied phenomenon.

7. You should be cautious in drawing conclusions because this study only includes young active males. Do not generalise your conclusion with your statement about using the autonomic flexibility index to assess individual CVS health.

Thank you very much for the question. I agreed with the review's comments. To avoid generalizing regarding our result, we describe in the discussion section, sub-section limitation (pages 18 and 19, lines 393 – 401): "Limitations in our work include the exclusion of older subjects, athletes, and women of any age range, so the findings cannot be extrapolated to these groups of people. Also, these results cannot be extrapolated to men in general because our sample was relatively homogenous, and thus men with different clinical/functional conditions would present different responses than those observed in the present study. Additionally, our findings cannot be extrapolated for other workload and duration exercises or protocols like a cycle or arm ergometry, where the individuals sit during rest, exercise and recovery. Also, the participants performed an active recovery after exercise in the orthostatic position, which limits the comparison to similar conditions". 

Conclusion 

Lastly, low RHR average values (< 60 bpm) showed to be a cardiac autonomic "flexibility" index for assessing individual cardiovascular health in young, physically active men.

Therefore, these finds suggest that RHR may be used as a preliminary tool for decision making, considering its potential helpful information related to individuals' cardiac autonomic capacity without the expense of clinical exercise tests to infer the parasympathetic activity sympathovagal balance response in the heart.

---

## [Decision Letter · Decision Letter 1]

14 Sep 2022

PONE-D-22-17046R1Can resting heart rate explain the heart rate and parasympathetic responses during rest, exercise, and recovery?PLOS ONE

Dear Dr. Garcia,

Thank you for submitting your manuscript to PLOS ONE. After careful consideration, we feel that it has merit but does not fully meet PLOS ONE’s publication criteria as it currently stands. Therefore, we invite you to submit a revised version of the manuscript that addresses the points raised during the review process.

 Please submit your revised manuscript by Oct 29 2022 11:59PM. If you will need more time than this to complete your revisions, please reply to this message or contact the journal office at plosone@plos.org. Please include the following items when submitting your revised manuscript:A rebuttal letter that responds to each point raised by the academic editor and reviewer(s). You should upload this letter as a separate file labeled 'Response to Reviewers'.A marked-up copy of your manuscript that highlights changes made to the original version. You should upload this as a separate file labeled 'Revised Manuscript with Track Changes'.An unmarked version of your revised paper without tracked changes. You should upload this as a separate file labeled 'Manuscript'.If applicable, we recommend that you deposit your laboratory protocols in protocols.io to enhance the reproducibility of your results. Protocols.io assigns your protocol its own identifier (DOI) so that it can be cited independently in the future. For instructions see: https://journals.plos.org/plosone/s/submission-guidelines#loc-laboratory-protocols. Additionally, PLOS ONE offers an option for publishing peer-reviewed Lab Protocol articles, which describe protocols hosted on protocols.io. Read more information on sharing protocols at https://plos.org/protocols?utm_medium=editorial-email&utm_source=authorletters&utm_campaign=protocols.

We look forward to receiving your revised manuscript.

Kind regards,

Laurent Mourot

Section Editor

PLOS ONE

Journal Requirements:

Reviewers' comments:

Reviewer's Responses to Questions

**Comments to the Author**

1. If the authors have adequately addressed your comments raised in a previous round of review and you feel that this manuscript is now acceptable for publication, you may indicate that here to bypass the “Comments to the Author” section, enter your conflict of interest statement in the “Confidential to Editor” section, and submit your "Accept" recommendation.

Reviewer #1: (No Response)

Reviewer #2: All comments have been addressed

2. Is the manuscript technically sound, and do the data support the conclusions?

Reviewer #1: Partly

Reviewer #2: Yes

3. Has the statistical analysis been performed appropriately and rigorously? 

Reviewer #1: Yes

Reviewer #2: Yes

4. Have the authors made all data underlying the findings in their manuscript fully available?

Reviewer #1: Yes

Reviewer #2: Yes

5. Is the manuscript presented in an intelligible fashion and written in standard English?

Reviewer #1: Yes

Reviewer #2: Yes

6. Review Comments to the Author

Reviewer #1: Abstract

L28: please delete “average”, that is implicit (throughout the manuscript as well)

L31: delete “and HR” that is implicit too as you measured the RR intervals

Introduction

L85: please re-examine this sentence: first delete “average”, then it is not clear how the HR can be both the independent and the dependent variable at the same time. You stated that it is important to know the effect of resting HR on HR at rest, which does not make any sense. You may want to test the hypothesis that different values of HR at rest are associated with different values of HRV, and different exercise and post-exercise responses etc.

L94: please consider making your hypothesis clearer. In which direction do you expect these relationships to go? e.g. A lower RHR = a faster recovery or a slower recovery? Obviously based on the literature you have already mentioned

Methods:

How did you perform the regression analysis? Was the predictor variable RHR considered as a continuous variable? Or you coded the two groups (e.g. BG = 0, NG=1)? please consider the first option as the best way to perform a linear regression analysis. What are the advantages of using the mean HR difference between the two groups instead of the raw values of RHR? Please clarify.

Results

You may want to add the ES descriptor (e.g. small/trivial etc..) in the tables, that will help the reader interpret your results. I also suggest mentioning (and discussing) the size of the effects in the following paragraphs to help the reader throughout the long discussion.

Discussion

I appreciate the effort done by the authors in clearly presenting their results, statistical power, etc. but how these trivial-to-small differences between the two groups in terms of exercise/post-exercise recovery responses and such small portions of the variance explained by the regression model can be translated into something helpful? How does this help clinical practice?

I also wonder whether correlation analysis, rather than group analysis, could somehow help explain how resting, exercise and post-exercise responses are related. Why did you choose this kind of approach?

These two points need to be presented and discussed in the manuscript.

Reviewer #2: Thank you for addressing all the comments raised by us. The manuscript now has scientific merit for publication.

7. PLOS authors have the option to publish the peer review history of their article (what does this mean?). If published, this will include your full peer review and any attached files.

Reviewer #1: No

Reviewer #2: **Yes: **Zulkarnain Jaafar

---

## [Author Response · Author response to Decision Letter 1]

14 Oct 2022

Responses to the reviewers 

2022/10/14

Manuscript title: Can resting heart rate explain the heart rate and parasympathetic responses during rest, exercise, and recovery?

Manuscript ID: PONE-D-22-17046R1.

Authors: Giliard Lago Garcia, Luiz Guilherme Grossi Porto, Carlos Janssen Gomes da Cruz and Guilherme Eckhardt Molina

We thank the Editor-in-Chief of PLOS ONE for the opportunity to resubmit our paper. We are very grateful to the reviewers for their comments and suggestions, which improved our manuscript. As requested, we have highlighted edits to facilitate further review. We believe the responses and additions satisfactorily address the reviewers' comments and suggestions and hope you will agree. The edits have clarified several points and have resulted in an improved manuscript, which we hope is now ready for final acceptance.

# To Journal Requirements:

Please review your reference list to ensure that it is complete and correct. If you have cited papers that have been retracted, please include the rationale for doing so in the manuscript text or remove these references and replace them with relevant current references. Any changes to the reference list should be mentioned in the rebuttal letter that accompanies your revised manuscript. If you need to cite a retracted article, indicate the article’s retracted status in the References list and also include a citation and full reference for the retraction notice.

All references are correct.

# To the Reviewer 1

Abstract:

L28: please delete “average”, that is implicit (throughout the manuscript as well)

We adjusted as requested.

L31: delete “and HR” that is implicit too as you measured the RR intervals

We adjusted as requested.

Introduction:

L85: please re-examine this sentence: first delete “average”, then it is not clear how the HR can be both the independent and the dependent variable at the same time. 

You stated that it is important to know the effect of resting HR on HR at rest, which does not make any sense. You may want to test the hypothesis that different values of HR at rest are associated with different values of HRV, and different exercise and post-exercise responses etc.

We thank the Reviewer's question. To avoid duality (misinterpretation) considering RHR and HR at rest, we adjusted the sentence throughout the manuscript. 

# New phase in the Introduction, page 2, line 53 (highlighted)

…, evaluated in the supine position,…

# New phase in the Introduction, page 4, lines 84 - 87 (highlighted)

To the best of our knowledge, no studies have shown information on the effect of different RHR values, evaluated in the supine position, on HR dynamic and parasympathetic activity at rest, during the exercise test, and recovery.

Also, the RHR evaluated in the supine position is the only independent variable as we highlighted in the Material and Methods section, page 6, lines134 - 138: “To test our hypothesis, the subjects were allocated into two groups based on the supine RHR values. Those who presented bradycardia at rest (RHR < 60 bpm) were grouped in the bradycardia group (BG n = 20), and those who presented RHR values within the normal range (HR ≥ 60 bpm < 100 bpm) were allocated to the normal HR group (NG n = 20).”. Accordingly, all other measures of HR dynamics and parasympathetic activity assessed at rest, during the exercise test, and during recovery are dependent variables.

Thus, to the best of our knowledge, no studies have shown information on the effect of different RHR (independent variable) values on HR dynamics (dependent variable) and parasympathetic activity (dependent variable) at rest, during the exercise test, and recovery.

# New phase in the Introduction, page 4, lines 95 - 97 (highlighted)

Therefore, we hypothesize that physically active young men with different RHR values in the supine position can show different HR dynamics and parasympathetic activity at rest, during, and after a submaximal exercise test.

L94: please consider making your hypothesis clearer. In which direction do you expect these relationships to go? e.g. A lower RHR = a faster recovery or a slower recovery? Obviously based on the literature you have already mentioned

We appreciate the Reviewer's question. However, we would like to highlight that this paper is not a correlational study but a comparative study. Thus, as we described in the Material and Methods section, page 10, line 227: “The two-tailed level of statistical significance was set at a p ≤ 0.05.” Therefore, our null hypothesis was that physically active young men with different RHR values in the supine position show the same HR dynamic and parasympathetic activity at rest, during, and after a submaximal exercise test, and our alternative hypothesis was that physically active young men with different RHR values in the supine position can show different HR dynamics and parasympathetic activity at rest, during, and after a submaximal exercise test.

# New phase in the Introduction, page 4, lines 95 - 97 (highlighted)

Therefore, we hypothesize that physically active young men with different RHR values in the supine position can show different HR dynamics and parasympathetic activity at rest, during, and after a submaximal exercise test.

Methods:

How did you perform the regression analysis? Was the predictor variable RHR considered as a continuous variable? Or you coded the two groups (e.g. BG = 0, NG=1)? please consider the first option as the best way to perform a linear regression analysis. What are the advantages of using the mean HR difference between the two groups instead of the raw values of RHR? Please clarify.

We are thankful to the Reviewer for this plausible and interesting question. We did not consider the predictor variable RHR (independent variable) as a continuous variable. We considered the predictor variable RHR as a dummy variable (BG = 0, NG=1) because we aim to develop an explication regression based on the effect of RHR values on the HR dynamics response during rest, exercise, and post-submaximal exercise in physically active young men, according to Shmueli, 2010 which describe the explication regression.

The predictive models tend to have higher predictive accuracy than explanatory statistical models; they can indicate the potential level of predictability. A very low predictability level can lead to the development of new measures, newly collected data, and new empirical approaches. An explanatory model close to the predictive benchmark may suggest that our understanding of that phenomenon can only be increased marginally. 

On the other hand, the explanatory model advantage is close to the predictive benchmark. Thus, the explanatory model would imply substantial practical and theoretical gains in further scientific development. Hence, explanatory modeling is commonly used for theory building and testing.

Shmueli, Galit. "To explain or to predict?." Statistical science 25.3 (2010): 289-310.

Results:

You may want to add the ES descriptor (e.g. small/trivial etc..) in the tables, that will help the reader interpret your results. I also suggest mentioning (and discussing) the size of the effects in the following paragraphs to help the reader throughout the long discussion.

Thank you very much for the question. We add the effect size descriptor in the tables as requested. We added the effect size descriptor as requested above by the reviewer.

Discussion:

I appreciate the effort done by the authors in clearly presenting their results, statistical power, etc. but how these trivial-to-small differences between the two groups in terms of exercise/post-exercise recovery responses and such small portions of the variance explained by the regression model can be translated into something helpful? How does this help clinical practice?

This is an excellent question. Thank you! According to described in the article (page 13, lines 270 – 275): " The NG presents on average 8.2 bpm in HRort and 6.2 bpm in HRinitial more than the BG, in which the RHR explains 17% and 22%, respectively, of the variance of this difference average between the groups. NG presents on average 5.5 % in ∆%RHR, 10.7 bpm in CR, 4.9 bpm in HRR 3rdmin, 6.4 bpm in HRR 5thmin, 3.1 bpm in %HRR3rdmin, and 3.9 bpm in %HRR5thmin lower than the BG, in which RHR explain from 11% to 48% o the variance of this difference average between the groups.".

Considering the present results of the percentage explained variance of difference average between the groups, we can say that RHR opens the possibility of a new approach (or analysis), which may add helpful information as a preliminary tool for decision-making in stress management for healthcare professionals bringing essential and complementary information related to individual cardiac autonomic capacity.

# New paragraph in the Discussion, page 17, lines 352 - 359 (highlighted)

Thus, the novelty of the present study is that these results may open a new possibility to the RHR (supine) analysis as a preliminary tool for decision-making, considering its potential helpful information related to the individual’s cardiac autonomic “flexibility” without the expense of clinical exercise tests to infer the parasympathetic activity on the heart in different functional conditions (rest, exercise, and recovery). From a clinical perspective, due to the presence of a small to large effect size, these results may be considered important to assess cardiovascular health, but this goes beyond our scope, and new studies should be done to confirm this hypothesis with the clinical population.

I also wonder whether correlation analysis, rather than group analysis, could somehow help explain how resting, exercise and post-exercise responses are related. Why did you choose this kind of approach?

Thank you very much for the question. First, the approach through correlation analysis can be a bit simplistic because it only presents the results of the direction and strength of the correlation. Thus, the correlation between the variables X and Y can arise either from X affecting Y or from Y affecting X. Sometimes X and Y can be cause and effect at the same time.

However, the simple linear regression analysis is more robust and requires that several assumptions are met to use it. Therefore, all variables utilized on simple linear regression were decided by the theoric, variables correlated with the independent variable, and only variables that presented significant statistics between groups because we have only one predictor variable (independent variable). 

All variables met the assumptions of normality, linearity of parameters, normality of residuals, independent values, homoscedasticity, absence of autocorrelation of residuals (Durbin-Watson test), absence of multicollinearity, and absence of outliers (Godfrey, 1985; Bewick, 2003).

Different from correlation analysis, simple linear regression not only test for relationships between variables but also quantifies their direction and strength. Thus, simple linear regression presented more robust measures than correlation analysis, such as the coefficient of determination (R2) and the regression coefficient (slope) (Hair, 2009).

Which, the coefficient of determination then represents the adjustment between the dependent variable (Y) and the independent predictor (X) variables that designate the effect. In other words, the coefficient of determination is the proportion of the variance of Y that can be predicted or explained based on its linear relationship with X. The stronger the coefficient of determination, the stronger the prediction or explanation of the association between the dependent variable Y and the independent variable X in linear terms. The minimum theoretical value of the coefficient of determination is 0% and the maximum is 100%, although the maximum is not expected in the health sciences (Espirito Santo, 2018). Also, remember that the coefficient of determination value is simply the squared correlation between the actual and predicted values. (Hair,2009).

The regression coefficient (slope) describes the average (expected) change in the dependent variable for each unit change in the independent variable. If the regression coefficient is perceived to be statistically significant (ie, the coefficient is significantly different from zero), the value of the regression coefficient indicates the extent to which the independent variable associates with the dependent one. (Hair, 2009)

Wherefore, according to described in the article (page 17, lines 360 – 363): “In addition, our findings may add important information based on a novel approach toward developing an explication regression based on the effect of RHR values on the HR response during these functional conditions (rest, exercise, and recovery) for a better understanding of these complex interactions”.

Bewick, Viv, Liz Cheek, and Jonathan Ball. "Statistics review 7: Correlation and regression." Critical care 7.6 (2003): 1-9.

Godfrey, Katherine. "Simple linear regression in medical research." New England Journal of Medicine 313.26 (1985): 1629-1636.

Hair, Joseph F. "Multivariate data analysis." (2009).

Espirito Santo, Helena, and Fernanda Daniel. "Calcular e apresentar tamanhos do efeito em trabalhos científicos (3): Guia para reportar os tamanhos do efeito para análises de regressão e ANOVAs [Calculating and reporting effect sizes on scientific papers (3): Guide to report regression models and ANOVA effect sizes]." Revista Portuguesa de Investigação Comportamental e Social 4.1 (2018): 43-60.

These two points need to be presented and discussed in the manuscript.

# New paragraph in the Conclusion, page 20, lines 422 - 426 (highlighted)

Therefore, we suggest that RHR (supine) analysis as a preliminary tool for decision-making, considering its potential helpful information related to the individual’s cardiac autonomic “flexibility” without the expense of clinical exercise tests to infer the parasympathetic activity on the heart in different functional conditions (rest, exercise, and recovery).

---

## [Decision Letter · Decision Letter 2]

4 Nov 2022

Can resting heart rate explain the heart rate and parasympathetic responses during rest, exercise, and recovery?

PONE-D-22-17046R2

Dear Dr. Garcia,

We’re pleased to inform you that your manuscript has been judged scientifically suitable for publication and will be formally accepted for publication once it meets all outstanding technical requirements.

Kind regards,

Laurent Mourot

Section Editor

PLOS ONE

Additional Editor Comments (optional):

Reviewers' comments:

Reviewer's Responses to Questions

**Comments to the Author**

1. If the authors have adequately addressed your comments raised in a previous round of review and you feel that this manuscript is now acceptable for publication, you may indicate that here to bypass the “Comments to the Author” section, enter your conflict of interest statement in the “Confidential to Editor” section, and submit your "Accept" recommendation.

Reviewer #1: All comments have been addressed

2. Is the manuscript technically sound, and do the data support the conclusions?

Reviewer #1: Yes

3. Has the statistical analysis been performed appropriately and rigorously? 

Reviewer #1: Yes

4. Have the authors made all data underlying the findings in their manuscript fully available?

Reviewer #1: Yes

5. Is the manuscript presented in an intelligible fashion and written in standard English?

Reviewer #1: Yes

6. Review Comments to the Author

Reviewer #1: (No Response)

7. PLOS authors have the option to publish the peer review history of their article (what does this mean?). If published, this will include your full peer review and any attached files.

Reviewer #1: No

---

## [Editor Report · Acceptance letter]

10 Nov 2022

PONE-D-22-17046R2 

Can resting heart rate explain the heart rate and parasympathetic responses during rest, exercise, and recovery? 

Dear Dr. Garcia:

I'm pleased to inform you that your manuscript has been deemed suitable for publication in PLOS ONE. Congratulations! Your manuscript is now with our production department. 

Kind regards, 

on behalf of

Dr Laurent Mourot 

Section Editor

PLOS ONE